# Application of an Eddy Current-Tuned Mass Damper to Vibration Mitigation of Offshore Wind Turbines

**Jijian Lian, Yue Zhao, Chong Lian, Haijun Wang \*, Xiaofeng Dong, Qi Jiang, Huan Zhou and Junni Jiang**

State Key Laboratory of Hydraulic Engineering Simulation and Safety, Tianjin University, No. 135 Yaguan Road, Jinnan District, Tianjin 300350, China; jjlian@tju.edu.cn (J.L.); yue_zhao@tju.edu.cn (Y.Z.); lianchongtju@126.com (C.L.); tju_skl@126.com (X.D.); jiangqi_tju@163.com (Q.J.); zhouhuan@tju.edu.cn (H.Z.); jiangjunni627@163.com (J.J.)
\* Correspondence: bookwhj@tju.edu.cn; Tel.: +86-22-2740-1123

**Abstract:** Offshore wind turbine (OWT) structures are highly sensitive to complex ambient excitations, especially extreme winds. To mitigate the vibrations of OWT structures under windstorm or typhoon conditions, a new eddy current with tuned mass damper (EC-TMD) system that combines the advantages of the eddy current damper and the tuned mass damper is proposed to install at the top of them. In the present study, the electromagnetic theory is applied to estimate the damping feature of the eddy current within the EC-TMD system. Then, the effectiveness of the EC-TMD system for vibration mitigation is demonstrated by small-scale tests. Furthermore, the EC-TMD system is used to alleviate structural vibrations of the OWT supported by composite bucket foundations (CBF) under extreme winds at the Xiangshui Wind Farm of China. It is found that the damping of the EC-TMD system can be ideally treated as having linear viscous damping characteristics, which are influenced by the gaps between the permanent magnets and the conductive materials as well as the permanent magnet layouts. Meanwhile, the RMS values of displacements of the OWT structure can be mitigated by 16% to 28%, and the acceleration can also be reduced significantly. Therefore, the excellent vibration-reducing performance of the EC-TMD system is confirmed, which provides meaningful guidance for application in the practical engineering of OWTs.

**Keywords:** offshore wind turbine; eddy current tuned mass damper; extreme winds; vibration mitigation; small scale test; prototype observation; numerical simulation

## 1. Introduction

Due to the increasing demand for electrical power and the decreases in fossil resources, most countries in the world have urgently turned to renewable and sustainable energy [1,2]. In the past decade, wind power, especially offshore wind power, has developed rapidly and become a significant source of renewable energy to supply electricity throughout the world [3]. By the end of the year 2017 [4], the total installed capacity of wind power has reached 513.5 GW, which is about 4.4 times as large as the capacity of 115.4 GW in 2008. Meanwhile, the total installed capacity of offshore wind energy has risen sharply from 1.4 GW to 18.7 GW during the past ten years with an average annual growth rate of 30% as shown in Figure 1. Compared with onshore wind power, offshore wind power shows many outstanding advantages such as better quality of wind resources, larger suitable free area to install, closer to the electricity consumption center and less influence on the environment [5,6]. Moreover, with the continuous development of offshore wind power, taller tower, longer blades and larger capacity generators will be recommended for use in wind farms to capture more wind resources and reduce unit production costs [7].

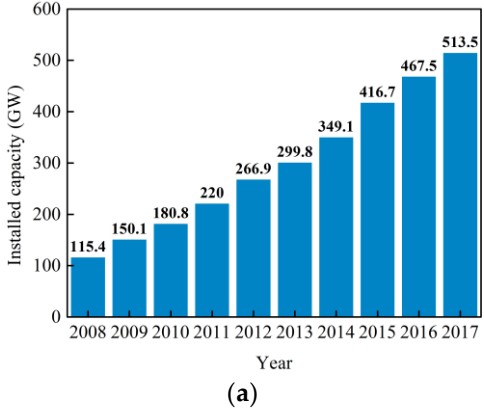 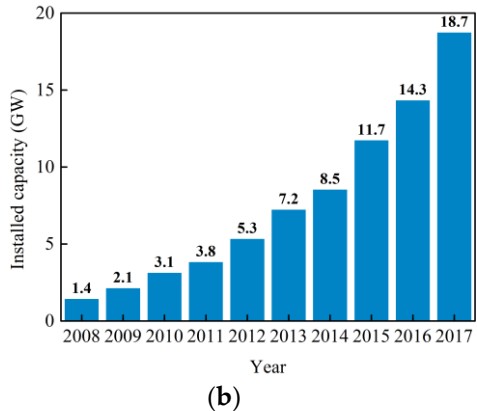

**Figure 1.** The installed capacity of worldwide wind power from 2007 to 2017: (**a**) Installed capacity of total wind power; (**b**) Installed capacity of offshore wind power [4].

China, with a mainland coastline of more than 18,000 km in length, has particularly rich offshore wind resources, which play important roles in accelerating the progress of green and renewable energy supply [7]. Nevertheless, the development of offshore wind power can face extreme winds. Being located in the west coast of the Pacific Ocean subjected to tropical anticyclones and the South China Sea monsoons [8], offshore wind farms in China experience super-typhoons annually, such as Maria (2018, 48 m/s), Hato (2017, 48 m/s), Meranti (2016, 45 m/s), Mujigae (2014, 50 m/s) and Rammasun (2014, 55 m/s). These extreme winds exert destructive power onto the offshore wind turbine (OWT) structures, causing great vibrations resulting in structural damage or even in collapses. In this regard, this can lead to a substantial economic loss or reduction in power production during the OWTs' service lifetime. Thus, it is meaningful to mitigate the dynamic responses of the OWT under extreme winds to ensure its structural safety and increase the reliability. To solve this problem, various types of vibration control devices and control methods have been proposed [9,10]. For instance, passive dampers, like tuned mass dampers (TMDs) [11–14], tuned (column) liquid dampers (TLDs or TCLDs) [15,16] and magneto-rheological fluid dampers (MRFDs) [17], have been extensively adopted to alleviate the vibration of OWTs and improve their resistance to multi-hazards. Among all the passive control devices, traditional TMDs are most widely used and feasible for vibration control [13,14]. For absorbing vibration energy and reducing dynamic responses, the natural frequencies of the TMDs should be set to be close to that of the controlled structures by adjusting the ratio between mass and spring or the length of the pendulum. However, viscous dampers employed in the TMDs have well-known problems that are possible leakage of oil or gas and difficulty in adjustment and maintenance [18]. Alternatively, an eddy current damper (ECD), as a contactless damper without modifying the stiffness of the controlled structures, is seen to be a potential and innovational solution to the above problems [18,19]. Moreover, the ECD has good energy dissipation ability, stable performance and reasonable robustness, which makes full use of the electromagnetic or eddy current damping forces [20].

The damping mechanism of the ECD was demonstrated by analytical models and experimental model tests. Wang et al. [18] conducted theoretical and experimental studies on a large-scale TMD with the ECD, indicating that the feasibility and reliability of the EC-TMD system in vibration mitigation. Sodano et al. [21,22] provided an improved mathematic model for predicting the damping force induced by the relative movement of the conductive materials in the magnetic fields, confirming a good performance of reducing the vibration responses of a beam. Ebrahimi et al. [23] analyzed the damping characteristics in the ECD through experiments and theory deviations and illustrated that it was applicable in vibration suppression systems. Bae et al. [24] applied a lightweight TMD with the ECD to a large beam structure, showing that the efficiency of the TMD in vibration attenuation was increased using the ECD. Lu et al. [25] estimated the effectiveness and performance of the EC-TMD system by installing it on the top of a steel-frame model and performing shaking table tests,

thus demonstrating the EC-TMD system was an excellent passive device forin suppressing vibrations under seismic excitation condtitions. Irazu et al. [26] analyzed the damping characteristics of the eddy current and proposed a new inverse method to simulate its influences on mitigating the vibrations of a cantilever beam. Many researchers have tried to utilize EC-TMD systems to solve practical engineering problems in long-span structures and high-rise buildings. Lei et al. [27] developed a new type of EC-TMD system and applied it into the practical engineering of the Rongjiang hangers of steel arch bridges to increase the damping ratio of the system. Chen et al. [19] utilized a new EC-TMD system including a permanent magnet plane for the vibration control of a large floor structure made of steel-concrete, improving the anti-vibration performance. Saige et al. [28] tried to improve the TMD effectiveness in reducing vibration induced by pedestrians on footbridges by adding eddy current damping. Lu et al. [29] performed different scale-model tests of anthe EC-TMD system, which was firstly used in the ultra-highrise building of the Shanghai Center Tower, noticeably attenuating the structural acceleration and displacement caused by winds and earthquakes.

The main contribution of this study is to exploit the effectiveness of the EC-TMD system in vibration control by experimental tests and apply it to the practical engineering of the OWT structures. To the best knowledge of the authors, it is the first time that this new type of the EC-TMD system is proposed to mitigate the vibration of the OWT structure under extreme winds. This paper is organized as follows: firstly, the damping mechanics of the EC-TMD system and its implementation in analytical models are introduced in Section 2. Then, experimental studies of the EC-TMD system in vibration mitigation are performed in small-scale model tests by free attenuation and base-excitation in Section 3. Additionally, the project overview and prototype observation of the CBF-supported OWT structures at the Xiangshui Wind Farm of China are introduced in Section 4. Section 5 contains numerical simulations for the EC-TMD system in improving resistance to t OWT structure vibration under extreme winds. Lastly, the conclusions of the whole work and the outlook for further studies are described in Section 6.

## 2. Vibration Mitigation of the EC-TMD System

### 2.1. EC-TMD System

Figure 2 illustrates a schematic diagram of the EC-TMD system and eddy current damping force. In this study, the EC-TMD system consists of a mass block, four steel cables, several permanent magnets (PMs), a copper plate (a conductive metal) and a steel plate located under the magnetic fields, as shown in Figure 2a.

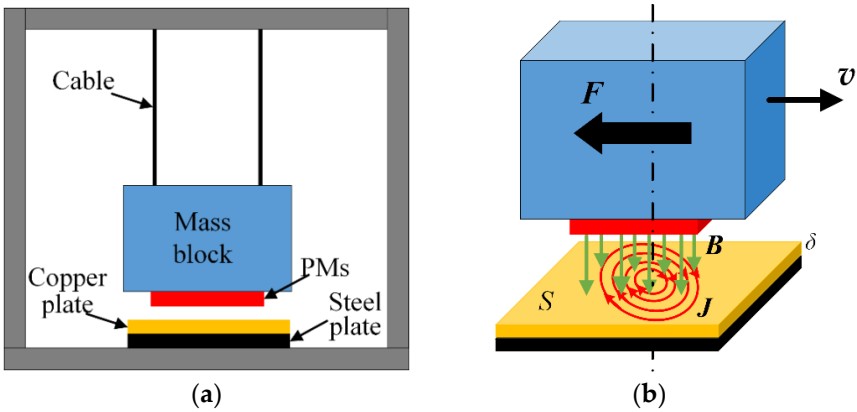

**Figure 2.** Damping mechanics of the ECD system: (**a**) Scheme of the EC-TMD system; (**b**) Eddy current damping force.

In Figure 2b, the black and green arrows depict the direction of the relative velocity $v$ and the magnetic induction intensity $B$, respectively, while the red arrows represent the direction of the eddy

current. In electromagnetic theory, when there is a relative velocity between the copper plate and the PMs, the eddy current is induced such that the interaction between the eddy current and magnetic fields will impede the relative movement [18]. In this regard, the eddy current damping effect or force can be generated. Therefore, the damping characteristics of the EC-TMD system can be contactless and adjustable through changing the material parameters or PM layouts within the system [18].

Assuming that the induced electric current density $J$ is only related with the conductive coefficient $\sigma$ of the copper plate, the relative velocity $v$ between the magnetic fields to the copper plate and the magnetic induction intensity $B$, neglecting the charge movement within the copper plate, which can be expressed as [21,29]:

$$J = \sigma(v \times B) \tag{1}$$

Additionally, the velocity $v$ and magnetic induction intensity $B$ can be expressed in vector form as [18]:

$$v = v_x i + v_y j + v_z k \tag{2}$$

$$B = B_x i + B_y j + B_z k \tag{3}$$

where $v_x$, $v_y$ and $v_z$ are the velocity $v$ in three directions, while $B_x$, $B_y$ and $B_z$ are the magnetic induction density $B$ in three directions, respectively. Assuming that the velocity $v$ is parallel to the $y$-direction and the $B$ is spatially distributed. Therefore, the $J$ can be expressed as:

$$J = \sigma v_y (B_z i - B_x k) \tag{4}$$

Then, the eddy current force $F$ related to the volume $V$ of the copper plate and the induced current density $J$ can be obtained as:

$$F = \int_V J \times B dV = \sigma v_y \int_V \left[ (B_x B_y) i - \left( B_x^2 + B_z^2 \right) j + (B_z B_y) k \right] dV \tag{5}$$

Hence, the eddy current force $F$ induced by the eddy current is given by:

$$F = -\sigma v_y \int_V \left( B_x^2 + B_z^2 \right) dV \tag{6}$$

In practical use, the movement of the copper plate is assumed perpendicular to the magnetic fields. Thus, the eddy current force $F$ can be described as [18]:

$$F = -\sigma \delta S B^2 v \tag{7}$$

where $\sigma$ and $S$ are respectively thickness and surface area of the copper plate, $v$ represents the relative velocity, and the negative sign indicates that the eddy current force is in the exactly opposite direction of the copper plate velocity. Under ideal conditions, the eddy current damping force is related with the relative velocity, indicating a linear viscous damping characteristic.

## 2.2. EC-TMD System in SDOF

For the sake of simplicity, the EC-TMD system is assumed to be installed at the top of a primary structure (single-degree of freedom, SDOF) constituting a two-degree of freedom system depicted in Figure 3. By considering the dynamic equilibrium condition, the equation of motion is presented as follows:

$$\begin{cases} m_s \ddot{x}_s + c_s \dot{x}_s + k_s x_s - c_d \left( \dot{x}_d - \dot{x}_s \right) - k_d (x_d - x_s) = P(t) \\ m_d \ddot{x}_d + c_d \left( \dot{x}_d - \dot{x}_s \right) + k_d (x_d - x_s) = 0 \end{cases} \tag{8}$$

where $m_i$, $c_i$ and $k_i$ are the mass, damping and stiffness of the primary structure ($i = s$) and the EC-TMD system ($i = d$), respectively. $\ddot{x}_i$, $\dot{x}_i$ and $x_i$ are the displacement, velocity and acceleration of the primary structure ($i = s$) and the EC-TMD ($i = d$), respectively. It is worth mentioning that the damping force

caused by the eddy current is nonlinear due to the unsteady changes in the gaps between the PMs and the copper plate. However, in most studies and practices, this eddy current damping force can be assumed to have ideal linear damping force characteristics, which will be adopted in the subsequent sections. In general, the mass ratio for TMD to the primary structure can justifiably range from 1% to 5% for taking account of efficiency and reliability [13,30]. Moreover, for a primary structure with light damping subjected to random white excitation, the optimal frequency related with the natural frequency of the primary structure and damping ratio of the EC-TMD system can be given by [31]:

$$\begin{cases} f_{opt} = f_1 \dfrac{(1+\mu/2)^{1/2}}{1+\mu} \\ \zeta_{opt} = \sqrt{\dfrac{\mu(1+3\mu/4)}{4(1+\mu)(1-\mu/2)}} \end{cases} \tag{9}$$

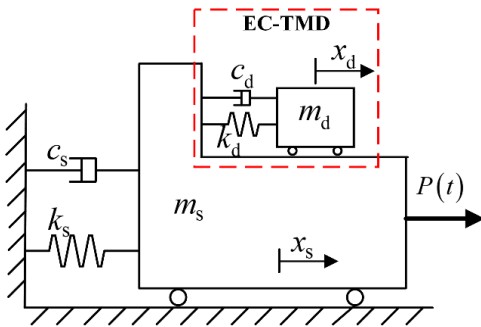

**Figure 3.** An equivalent two-degree of freedom system.

## 3. Experimental Study of the EC-TMD System in Vibration Mitigation

To validate the effectiveness of the EC-TMD system in reducing vibration responses, a series of small-scale model tests were conducted at the Hydraulic Power Center of Tianjin University, China. Unlike many previous studies that EC-TMD system was investigated alone, the EC-TMD system is installed on the top of a primary structure. The gaps between the PMs and the copper plate and the PMs layouts are investigated on the influence of the damping ratio of the whole system.

Figure 4 describes the schematic diagram and real structures of the experiment consisting of a shaking table and control system, a primary structure and the EC-TMD system, and an acquisition system, in which 'a' represents acceleration sensor and 'd' represents vibration displacement sensor.

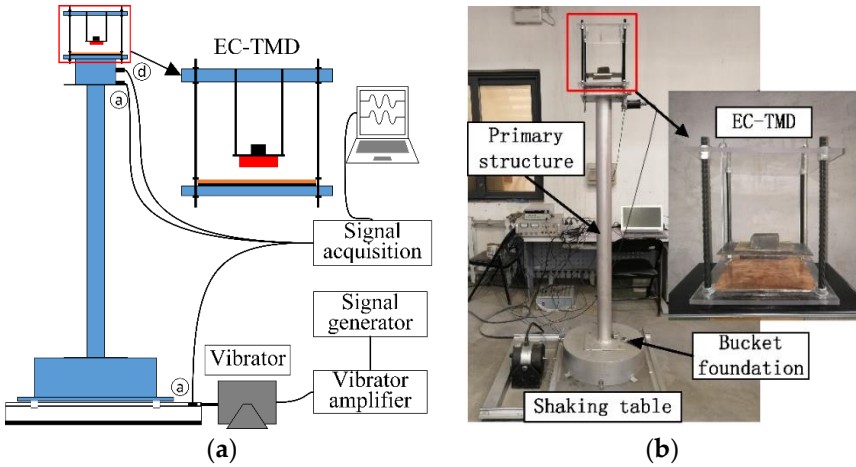

(**a**)          (**b**)

**Figure 4.** Experimental layouts of the EC-TMD system: (**a**) Layouts of the test; (**b**) Real structure of the test.

### 3.1. The Primary Structure

The primary structure, which mainly comprises a scale model (1:20) of a bucket foundation, a tube and a mass block, is made of steel. Table 1 gives the properties for the primary structure. As a supporting part, both of the bottom and top steel plate have $240 \times 240 \times 5$ mm dimensions, and the bottom one is fixed to a bucket foundation by four bolts then fixed to a shaking table by six bolts.

**Table 1.** Properties of the primary structure.

| Column Length (mm) | Section (mm) | Mass Block (kg) | Bucket Foundation (kg) | Theoretical Natural Frequency (Hz) |
|---|---|---|---|---|
| 1500 | D 80, d 76 | 7.85 | 6.78 | 2.3 |

### 3.2. Parameters of the EC-TMD System

Figure 4 describes the layouts of the EC-TMD system. According to the references [18,32], $Nd_2Fe_{14}B$ is selected as the PMs material and the size of the PM is $0.1 \times 0.05 \times 0.01$ m, while the size of the copper plate is $0.2 \times 0.2 \times 0.002$ m. The mass of the TMD and its natural frequency are 2.0 kg and 2.2 Hz, respectively, which can be adjustable by changing the length of the cables. The TMD is suspended by the cables from a stiffness plexiglass plate. The PMs are attached to the bottom of the TMD mass block, while the copper plate and steel plate are fixed on the plexiglass plate. Supporting structures consisting of four screw rods are used to support the plexiglass plate and adjust the distance between the two plates.

### 3.3. Test Results and Discussions

The free attenuation curves of displacements with various gaps between the copper plate and the PMs are shown in Figure 5.

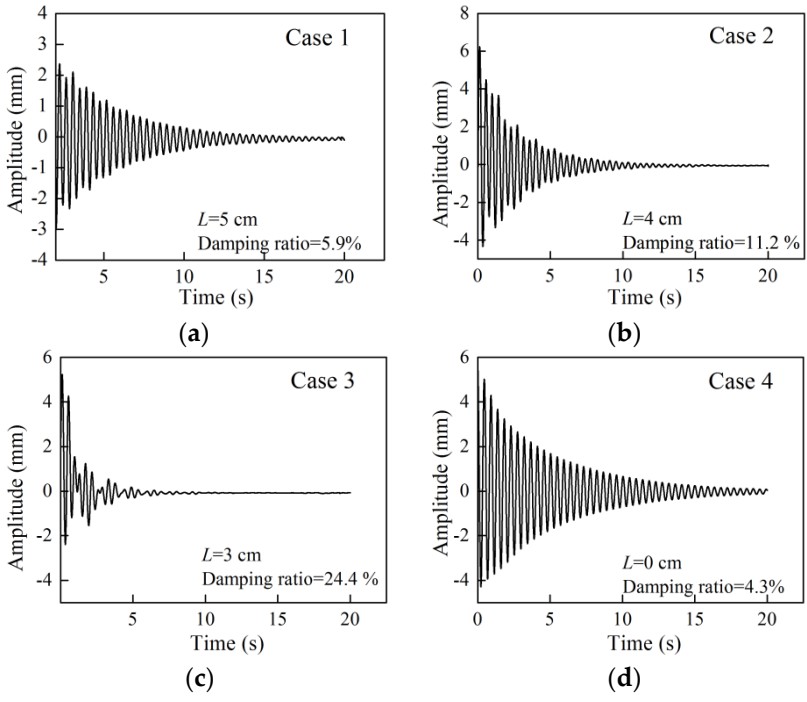

**Figure 5.** Free attenuation curves of the model tests: (**a**) Free attenuation of Case 1; (**b**) Free attenuation of Case 2; (**c**) Free attenuation of Case 3; (**d**) Free attenuation of Case 4.

The gaps (*L*) between the PMs and copper plate are chosen as 5 cm, 4 cm, 3 cm and 0 cm, respectively noted as Case 1, Case 2, Case 3 and Case 4, in which Case 4 indicates that there is no

additional damping effect of the EC-TMD system. Hence, the natural frequency and damping ratio of the primary structure are 2.2 Hz and 4.3%, respectively. In the four cases, the copper plate thickness and PMs layouts are kept the same. In this figure, Case 1 can be used to represent the EC-TMD system with little eddy current damping, while Cases 2 and 3 present increasingly more EC-TMD damping as the gaps decrease and the magnetic field increase in the copper plate. It is worth noting that free attenuation curve in Case 3 shows some unstable feature and then quickly return to stability. The reason for such a curve may be that nonlinear eddy current damping has a greater influence on the dynamic characteristics of the primary structure than the inertial force and gravity in the EC-TMD system. In the subsequent study, nonlinear eddy current damping should be avoided which is beyond the scope of this study.

　　Furthermore, subsequent tests were conducted to determine the influencing factors of the different parameters on the damping ratio of the EC-TMD system. The copper plate thickness is fixed at 2 mm. PMs layouts include three cases shown in Figure 6, while Figure 7 presents free attenuation curves for the first two PMs layouts. Furthermore, the results are collected in Table 2. It can be observed that the damping ratios of III PMs layout is larger than that of the first two PMs layouts due to stronger magnetic fields and eddy current damping forces, while there is the same of the damping ratio for the first two PMs layouts.

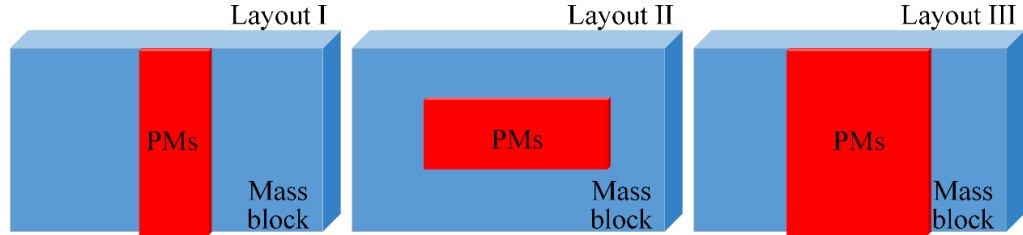

**Figure 6.** Three different PMs layouts.

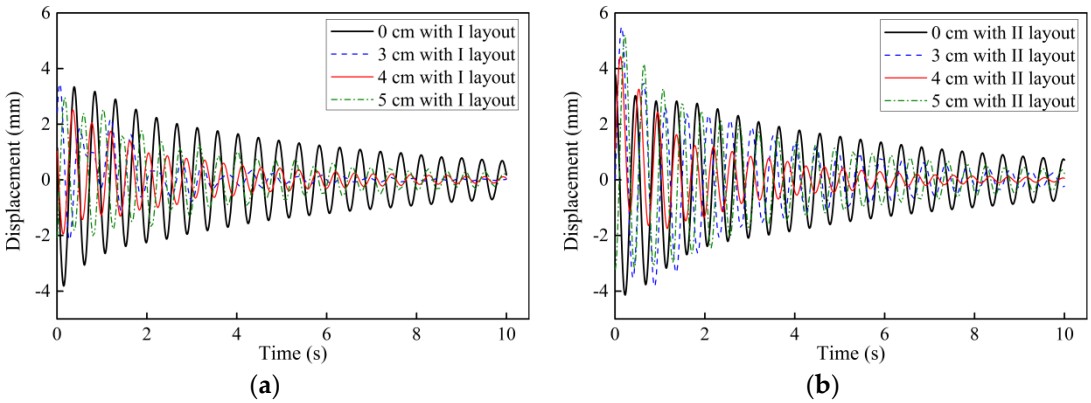

**Figure 7.** EC-TMD free attenuation curves of model tests: (**a**) Free attenuation curves of the PMs in I layout; (**b**) Free attenuation curves of the PMs in II layout.

**Table 2.** The damping ratio of the EC-TMD system with different configurations.

| PM Layouts | Gaps (cm) | | | |
|:---:|:---:|:---:|:---:|:---:|
| | **5** | **4** | **3** | **0** |
| I | 5.9% | 11.2% | 24.4% | 4.3% |
| II | 5.9% | 12.4% | 24.9% | 4.3% |
| III | 9.1% | 16.5% | 35.6% | 4.2% |

　　As to the small-scale model subjected to base-excitations caused by the shaking table, Figure 8 provides the displacement time histories at the top of the model under different base-excitations,

the gaps and PMs layouts. As shown, a forced vibration phenomenon is observed in this figure, in which the vibration response period is the same as that of the base-excitation. Meanwhile, the EC-TMD system is excited, and relative movement between the PMs and the copper plate occurs, resulting in the magnetic field changes within the copper plate and producing eddy current damping force. Moreover, in the same PMs layout, there are various vibration displacement amplitudes in different base-excitation frequencies. For instance, in the I and III PMs layouts conditions, the gaps for the minimum vibration amplitudes are 4 cm, 3 cm and 2 cm corresponding to the base-excitation frequencies of 1.5 Hz, 2.0 Hz and 2.5 Hz shown in dot lines in the figure. The possible reasons for these differences are attributed to the coupling interaction of the magnetic force and excitation force. To be specific, in the EC-TMD system, if the magnetic field force between the PMs and the steel plate is even stronger than the driving force of the system bottom, the relative movement between the PMs and the copper plate is reduced, so as to the eddy current damping force. On the contrary, if the magnetic field force is much smaller than the driving force, the relative motion of the PMs and the copper plate is enhanced, and the TMD in the EC-TMD system mainly exhibits its vibration damping characteristic, and the eddy current will add a damping force to the whole structure, too. Additionally, if the magnetic field force is equivalent to the driving force, both the eddy current damping and the TMD's damping work together to provide a reaction force and dissipate energy for the primary structure. Therefore, influencing factors as the gaps, the PMs layout and magnetic field intensity should be taken into consideration for proper application in practical engineering.

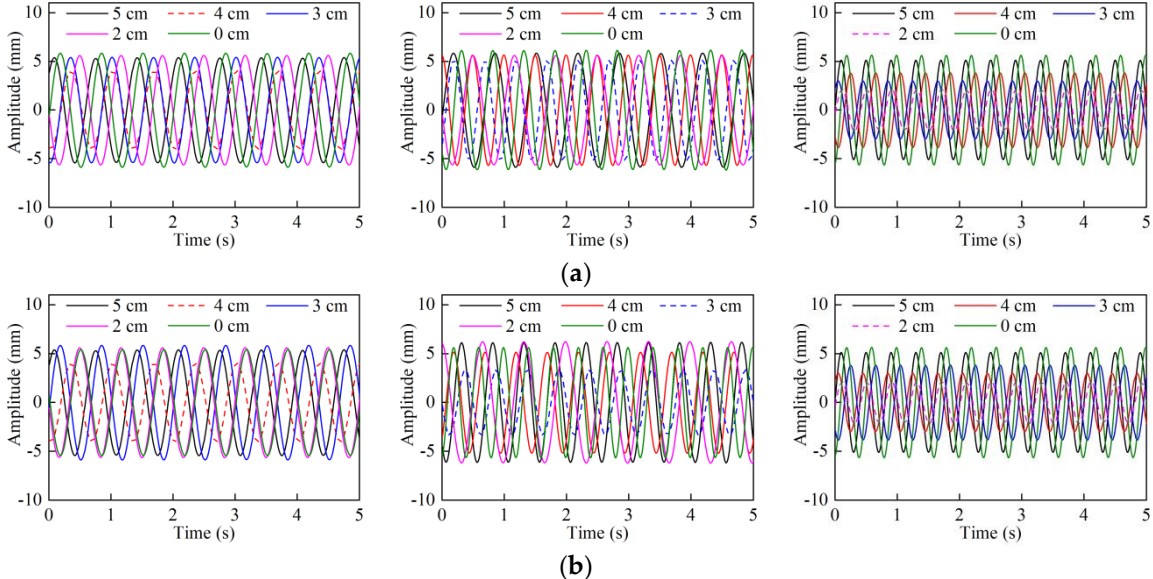

**Figure 8.** Displacement time histories under different base-frequency excitations: (**a**) Displacement time histories of the PMs in I layout; (**b**) Displacement time histories of the PMs in III layout.

## 4. Project Overview and Prototype Observation

The location of the Xiangshui Wind Farm and a photograph of the CBF are depicted in Figure 9. The OWT is supported by the CBF situated in the Yellow Sea areas belonging to Xiangshui County in Jiangsu Province of China. The offshore distance of the wind farm is about 10 km (the straight line distance between the wind farm center and the nearest shoreline), and the wind farm is about 10 km long and 2.5 km to 5.5 km wide perpendicular to the coastline with an area of about 90 km$^2$, where the planned installed capacity is 200 MW. The CBF consists of a steel bucket with 30.0 m diameter and 12 m height, pre-stressed concrete transition part with 5.1 m to 20 m diameter and 20 m height, which has a total weight of about 2700 t. The parameters of the 3.0 MW OWT can be found in Table 3.

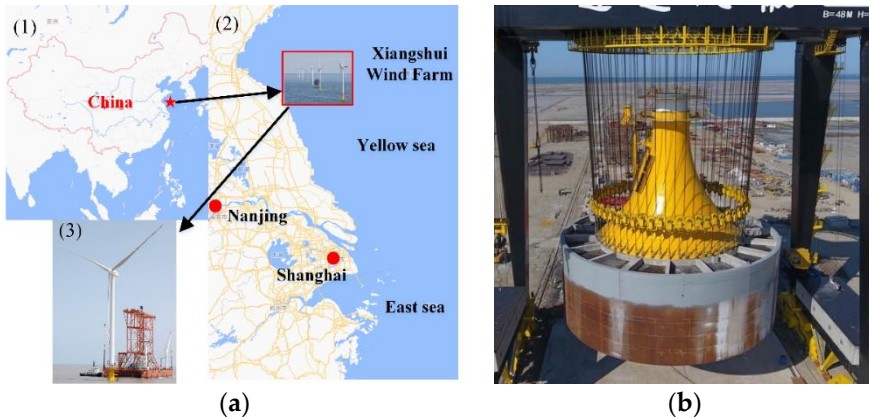

**Figure 9.** Composite bucket foundation in Xiangshui Wind Farm: (**a**) Location of Xiangshui Wind Farm; (**b**) Composite bucket foundation.

**Table 3.** Parameters of the 3.0 MW OWT.

| Parameters | Values |
|---|---|
| Power rating (MW) | 3.0 |
| Blade number | 3 |
| Hub height above sea level (m) | 90 |
| Tower diameter base, top (m) | 4.3, 3.2 (linear variation) |
| Tower thickness base, top (mm) | 50, 30 (linear variation) |
| Rotor-Nacelle mass (t) | 190.0 |
| Cut in, cut out wind speed (m/s) | 3, 25 |

An innovative one-step installation technology is used to install the OWT structure, which is assembled onshore, then floated to the target location by ship and finally installed at the designed depth by self-weight and negative pressure [33,34]. After installation of the CBF, the structural vibration response signals can be obtained by the prototype observation with several sensors installed at the top of the tower [35]. The X- and Z-directions are the tangential and radial direction of the tower wall in the same horizontal plane [35]. To obtain the dynamic characteristics of the OWT under parking conditions, the observed signals in X- and Z-directions are chosen to be analyzed with the mean wind speeds of 2.0 m/s and 3.1 m/s, and time histories and normalized power spectrum density (PSD) are presented in Figure 10. As shown, the corresponding frequencies to peak values of the vibration responses in X- and Z-directions are concentrated at about 0.35 Hz, which should be considered as the natural frequency of the OWT structure. Furthermore, the half-power bandwidth method [36] is implemented to obtain the damping ratio of the OWT structure. Thus, both the results of the natural frequencies and damping ratio can be employed in the subsequent numerical simulations, as shown in Table 4.

Moreover, under operational conditions, the vibration time histories in X- and Z-directions are displayed in Figure 11, with mean wind speeds of 4.1 m/s, 8.5 m/s, 16.1 m/s and 17.1 m/s. Thus, there are the resultant displacements apparently increase with the mean wind speeds, which is in line with the reference [35].

**Table 4.** Natural frequency and damping ratio of the OWT in parking conditions.

| Mean Wind Speed | $f_0$ | $f_1$ | $f_2$ | $\xi$ | Direction |
|---|---|---|---|---|---|
| 2.0 m/s | 0.348 | 0.338 | 0.352 | 2.2% | X-direction |
| | 0.350 | 0.339 | 0.352 | 2.0% | Z-direction |
| 3.1 m/s | 0.349 | 0.340 | 0.354 | 2.1% | X-direction |
| | 0.350 | 0.340 | 0.354 | 2.0% | Z-direction |

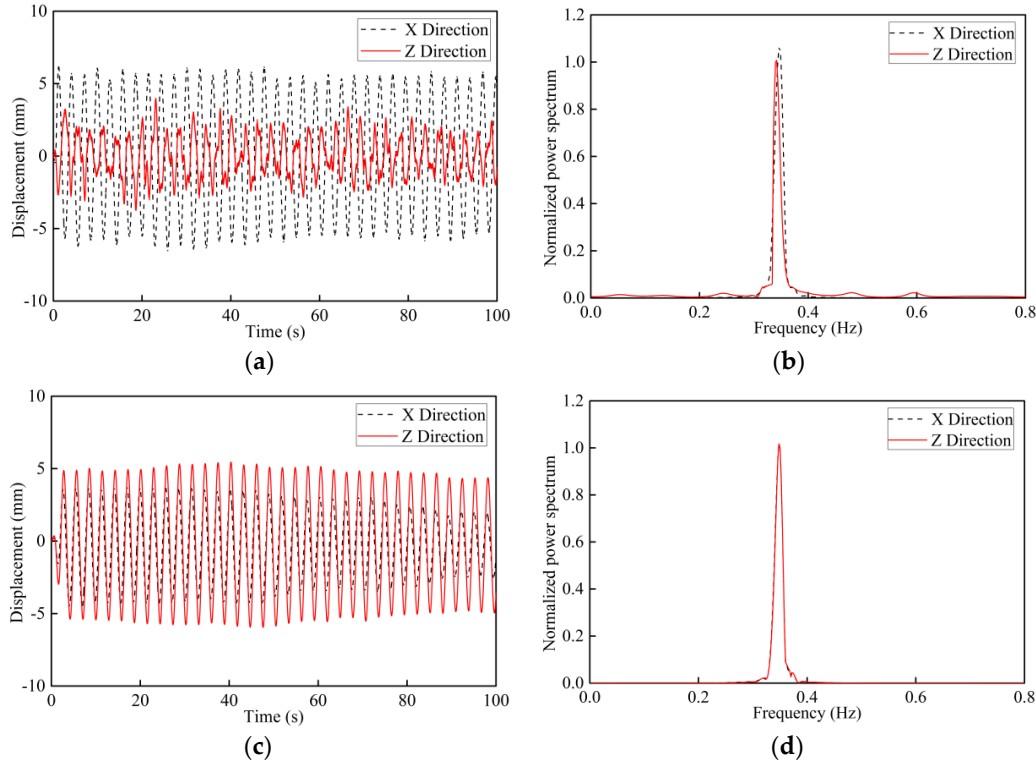

**Figure 10.** Observed data of the OWT in parking conditions: (**a**) Displacement time histories of 2.0 m/s; (**b**) Normalized PSD of 2.0 m/s; (**c**) Displacement time histories of 3.1 m/s; (**d**) Normalized PSD of 3.1 m/s.

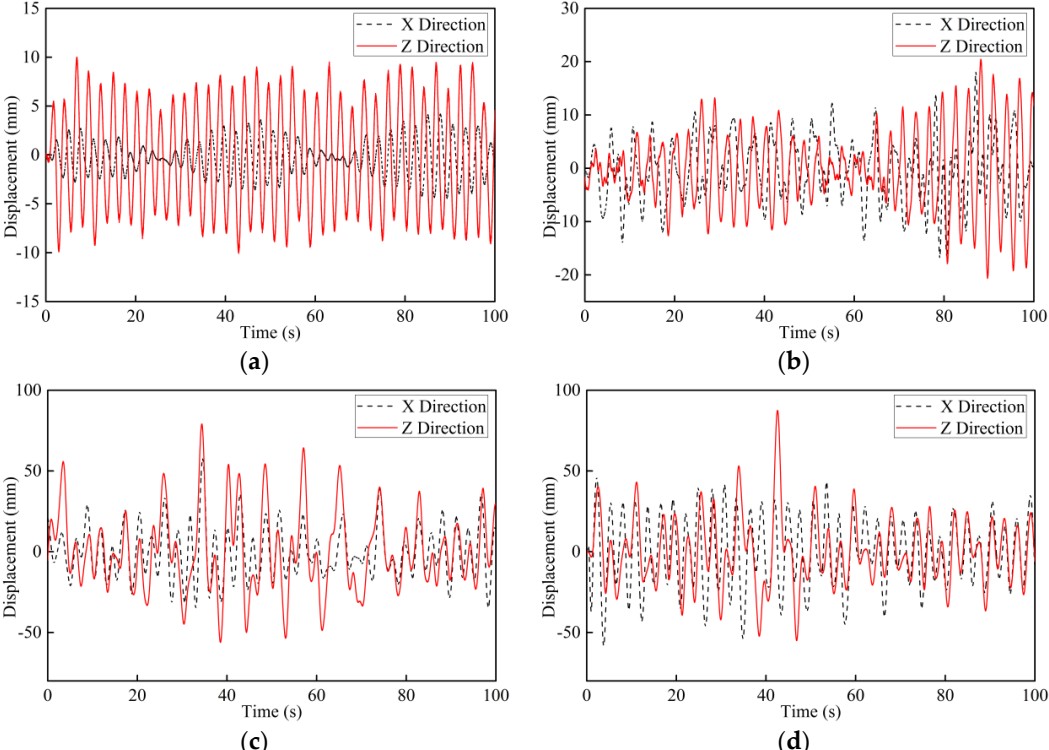

**Figure 11.** Displacement time histories in X- and Z-directions in operational conditions: (**a**) Displacement time histories of 4.1 m/s; (**b**) Displacement time histories of 8.5 m/s; (**c**) Displacement time histories of 16.1 m/s; (**d**) Displacement time histories of 17.1 m/s.

## 5. Numerical Simulation for the OWT Structure

### 5.1. Finite Element Models for the OWT and EC-TMD

Detailed 3D finite element models (FEM) generated by the by ABAQUS software are implemented to investigate the application of the EC-TMD system in the CBF-supported OWT structure under extreme winds. Figure 12 displays the 3D FEM of the OWT supported by the CBF considering the surrounding soil. For the sake of simplicity, a mass block set on the top of the tower can be used to represent the nacelle and blades neglecting the complex coupling between the rotors and the tower, and the soil bottom assumed to be fully constrained. The diameter $D$ of the CBF is 30 m and height $H$ is 10 m. To minimize the influences of the boundary effect, the radius and height of the soil are $5D$ and $6H$, respectively, and an infinite-element boundary is implemented in this study. Materials for the OWT structures as the tower, bucket foundation and transition piece are supposed to be elastic. However, the soil is divided into six layers modeled as an elastic-plastic model (Mohr-Coulomb criterion) [37]. Figure 12 contains the parameters for the layered soil including the thickness, effective unit weight, Young's modulus, friction angle and cohesive stress. In the model, 8-node linear brick elements with reduced integration (C3D8R) is used for the 3D solid elements, 2-node linear elements (T3D2) for the truss elements, infinite elements (CIN3D8) for the infinite-element boundary and 4-node doubly curved thin elements (S4R) for the shell elements. The surface-to-surface contact is applied to simulate the interaction between the bucket and the soil.

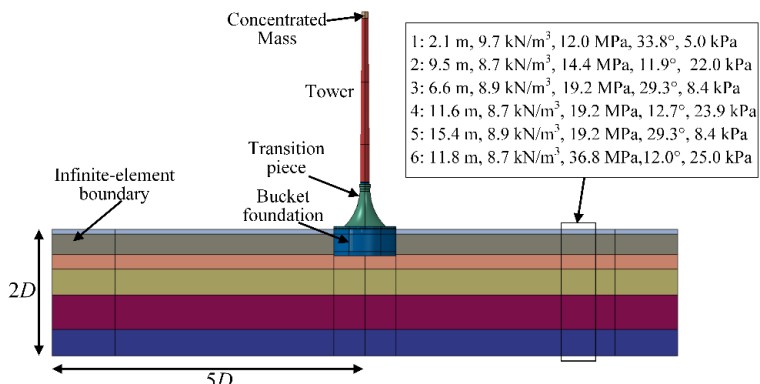

**Figure 12.** The half of the FEM model used in this study.

The mode shapes and natural frequencies of the OWT structure are acquired by implementing an eigenvalue analysis. As shown in Table 5, there are almost the same mode shapes and frequencies in the two horizontal directions for the identical mode order due to the symmetry of the FEM. In this study, the dynamic responses of the OWT structure can be mostly determined by the first and second vibration modes. The first natural frequency of the model is 0.348 Hz to 0.35 Hz, which is almost the same as the measured values in Table 4. Therefore, the simulated model can be used to represent the real OWT structure. In general, the damping ratio of the OWT structure is intricate, because there are different sources of damping as aerodynamics, hydrodynamics and materials. Nevertheless, according to the prototype observation analysis, the 2.0% damping ratio by prototype observation is adopted for the first vibration mode, and Rayleigh damping is considered in the numerical simulation. Therefore, the mass and stiffness coefficients are respectively calculated as 0.077 and 0.0022.

**Table 5.** Structural dynamic properties in ABAQUS eigenvalue analysis.

| Mode Order | Eigenvalue by ABAQUS | Model |
|:---:|:---:|:---:|
| 1st | 0.348 | For-aft |
| 1st | 0.350 | Side-side |
| 2nd | 2.521 | For-aft |
| 2nd | 2.525 | Side-side |

Table 6 tabulates the configuration parameters of EC-TMD system reasonably determined by the optimal equations [31]. In Table 6a, the conductive coefficient $\sigma$, the magnetic induce intensity $B$, the thickness $\delta$ and surface acre $S$ of the copper plate are given for two different eddy current dampers. As to the EC-TMD system, the mass ratio, frequency ratio, the eddy current damper and the length of cable are shown in Table 6b. In the 3D FEM, the extreme winds loads are assumed to act in the X-direction of the OWT structure. The EC-TMD devices are therefore applied in the X-direction, too. Figure 13 presents the schematic diagram and the FEM of the EC-TMD system, in which four cables and dashpots are used to suspend and connect a mass block and the tower, in which the dashpots stands for the equivalent eddy current damping. The blue disk stands for the copper plate and the yellow one for the PMs. According to the optimal damping, different numbers of the PMs are arranged as a disk under the mass block. The copper plate with a diameter of 2 m and a thickness of 4 mm is fixed to the working platform in the tower. As mentioned in the small-scale tests, the 4 cm is recommended to be the gap between the PMs and the copper plate. It is worth noting that the tolerable vibration displacement of the mass block is about 1.5 m due to the limitation of the tower space.

**Table 6.** Configuration parameters of EC-TMD system used in the OWT structure: (**a**) Configuration parameters of the eddy current damping; (**b**) Configuration parameters of the EC-TMD system.

| (a) | | | | |
|---|---|---|---|---|
| **Eddy Current** | **$\sigma/(1/\Omega \cdot m)$** | **B/T** | **$\delta$/mm** | **S/m$^2$** |
| EC 1 | $5.6 \times 10^7$ | 0.65 | 4 | 0.2 |
| EC 2 | $5.6 \times 10^7$ | 0.65 | 4 | 0.4 |

| (b) | | | | |
|---|---|---|---|---|
| **EC-TMD** | **Mass Ratio** | **Frequency Ratio** | **EC Structure** | **Length of Cable/m** |
| EC-TMD 1 | 0.03 | 0.98 | EC 1 | 2.12 |
| EC-TMD 2 | 0.05 | 0.96 | EC 2 | 2.18 |

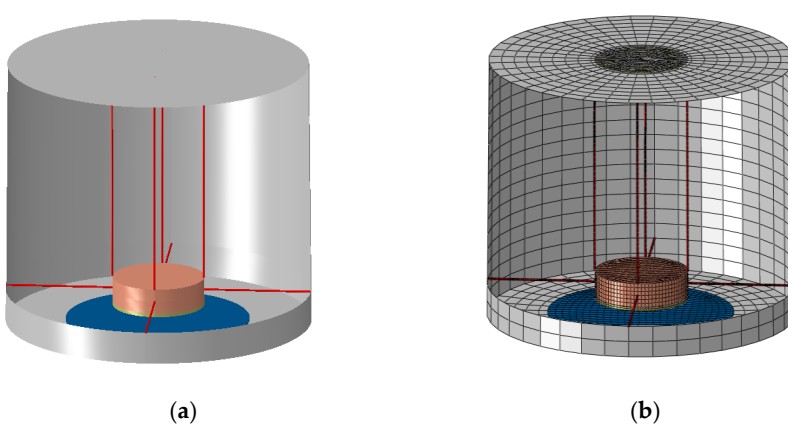

(**a**)　　　　　　　　　　　　　　　　　　　　(**b**)

**Figure 13.** The schematic diagram and the FEM of the EC-TMD system: (**a**) The schematic diagram of the EC-TMD; (**b**) The FEM of the EC-TMD system.

## 5.2. Extreme Wind Excitation

In the present study, the EC-TMD system is used to mitigate the dynamic responses of the OWT structure under extreme winds. The wind loading on the OWT can be normally decomposed into two parts caused by mean wind speeds and fluctuating wind speeds, respectively. Thus, a summation of the mean and the fluctuating components constitute the total wind force on the structure. According to the code [38], the spectral density of stochastic wind speeds can be represented by the Kaimal spectrum, which is widely used in practice and studies:

$$S_k(f) = \sigma_k^2 \frac{4L_k/V_{hub}}{(1 + 6fL_k/V_{hub})^{\frac{5}{3}}} \tag{10}$$

in which $f$ is the frequency, $S_k$ is the velocity component spectrum, $\sigma_k^2$ is the deviation of the wind speed component, $V_{hub}$ is the 10-min horizontal mean wind speed at the top of the tower, and $L_k$ is the velocity component integral scale parameter. For details about the fluctuating wind and fluctuating wind force denoted as $f_i(t)$, one may refer to [13]. The wind force induced by the mean wind speed can be calculated as:

$$\overline{f}_{mean,i} = \frac{1}{2}C_D A_i \rho \overline{v}_i^2 \tag{11}$$

where $\rho$, $C_D$ are the air density and the drag coefficient. $A_i$, $\overline{v}_i$ denote the projected areas to wind direction and the mean wind speed at the location $i$.

Therefore, the total wind force on the OWT structure at the different segments, shown in Figure 14, can be described as [13]:

$$F_i(t) = \overline{f}_{mean,\, i} + f_i(t) \tag{12}$$

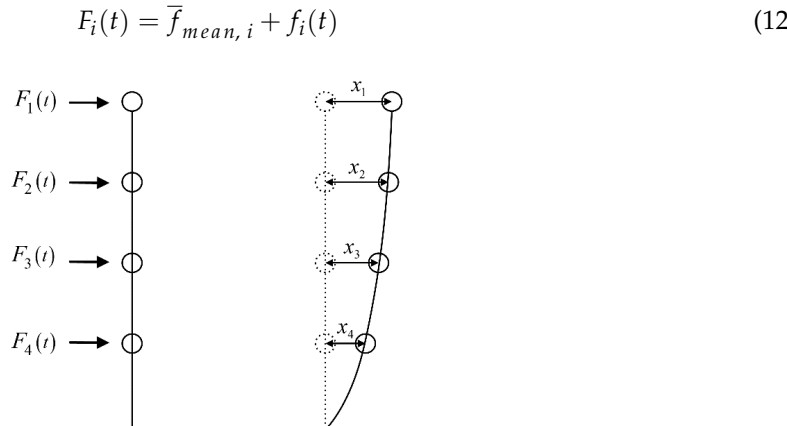

**Figure 14.** Wind force and displacement at different segments along the tower.

In the numerical simulation, the drag force coefficient and air density are 1.2, 1.225 kg/m$^3$, respectively. Furthermore, the turbulence extreme wind model [38] is considered to validate the performance of the EC-TMD system in vibration reduction in the OWT. Two recurrence periods of 1 year and 50 years are studied, in which the turbulence extreme wind model uses the 10-min mean wind speed at the top of the tower, and the reference wind speeds $\overline{v}$ are chosen to be 50 m/s (class I), 42.5 m/s (class II) and 37.5 m/s (class III) [38]. For instance, when the mean wind velocity at the top of the tower is taken as 37.5 m/s with 1 year's recurrence period. Figure 15 illustrates the comparisons between the simulated and the target PSD, and the simulated drag forces at four segments calculated by Equation (12).

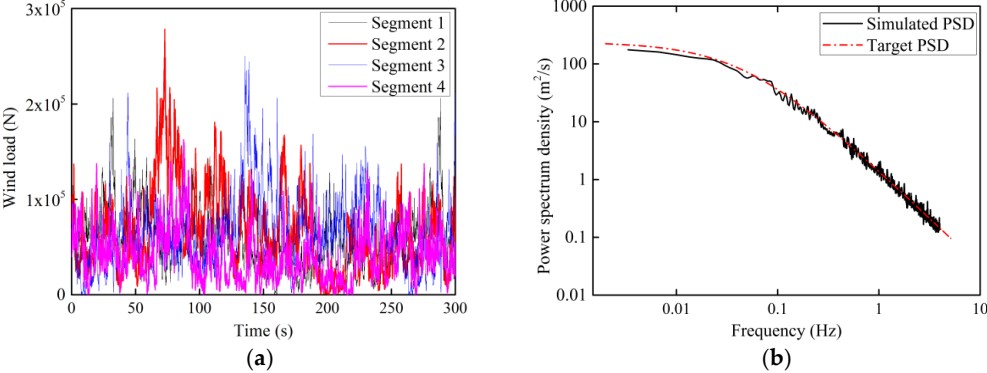

**Figure 15.** Simulated wind forces and the comparisons of the PSD with 37.5 m/s wind speed with 1 year's recurrence period: (**a**) Simulated wind forces; (**b**) Comparisons of the simulated and target PSD.

*5.3. Resutls*

To examine the feasibility of the EC-TMD system in reducing the dynamic responses induced by extreme winds, six extreme wind scenarios are considered, which are assumed to be the excitation sources applied to the OWT with a period of 300 s. The responses of the OWT structure with and without EC-TMD system are investigated and compared. Figure 16 depicts the displacement time histories at the top of the tower under extreme winds with Class I and Class III by 1 and 50 years' recurrence periods. As shown in the figure, during the initial stage of about 10 s, the EC-TMD system is not effective due to little relative movement between the system and the OWT structure. After that, the vibration-reducing performance of the EC-TMD system is quite noticeable that the displacements of the OWTs with the EC-TMD system are much smaller than those without the EC-TMD system, especially in the cases with 50 years' recurrence period. Moreover, there is a slightly better vibration-reducing performance of the EC-TMD 1 system than the EC-TMD 2 system. Nevertheless, in some periods, there are some similar vibration displacements for the conditions with and without EC-TMD system, because of the stochastic features of the extreme wind excitation resulting in the vibration synchronization of the EC-TMD system and the OWT structure. Table 7 collects the peak values of displacement at the tower top for these six cases, in which the vibration amplitudes can be reduced by about 23% to 35%, while the RMS values reduced by about 16% to 28% as shown in Table 8. Furthermore, it can be noticed that there are even more reduction percentage for double peak values (peak values minus valley values). In addition, the normalized PSD curves of the corresponding acceleration at the tower top can be observed in Figure 17. The peak values for no damper condition is normalized to one, and corresponding values in X-axis is about 0.35 Hz near to the natural frequency of the OWT structure. With the EC-TMD system, the acceleration PSD curves become wider and less peak values indicating an evident vibration-reducing effect [11,12].

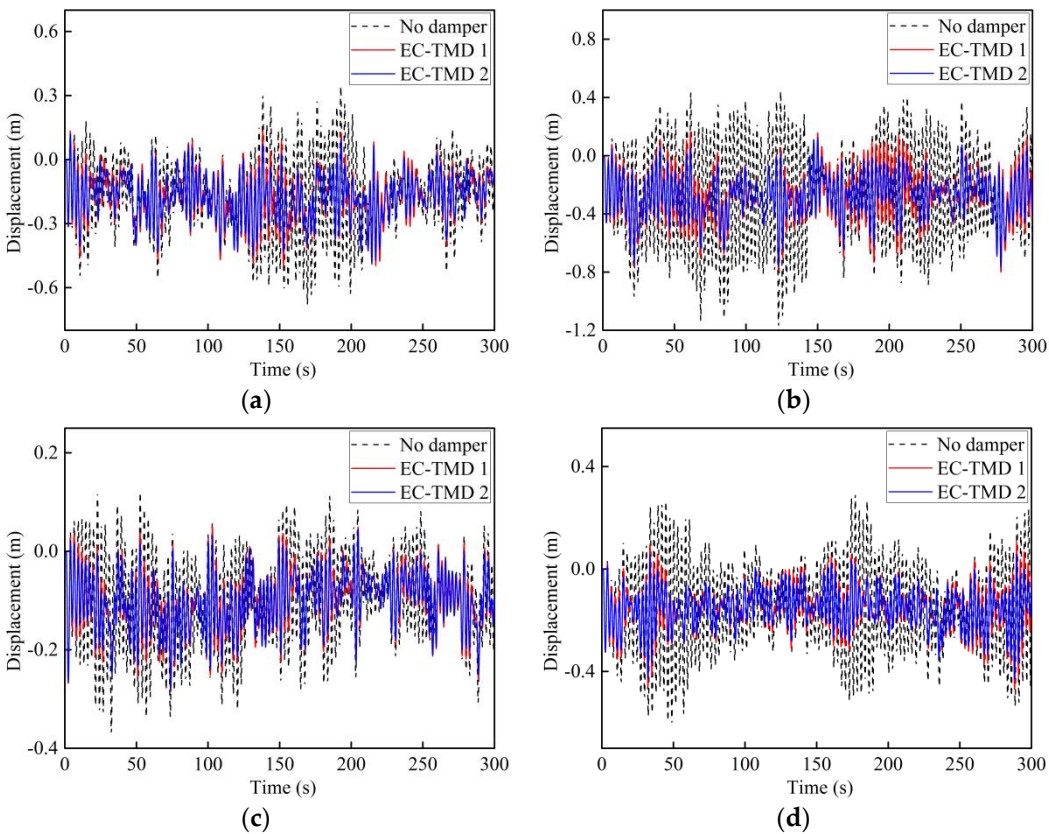

**Figure 16.** Displacement time histories with 1 and 50 years' recurrence periods: (**a**) Class I with 1 year's recurrence period; (**b**) Class I with 50 years' recurrence period; (**c**) Class III with 1 year's recurrence period; (**d**) Class III with 50 years' recurrence period.

**Table 7.** Peak values of displacements at the tower top for different cases.

| Cases | No Damper (m) | With EC-TMD 1 (m) | With EC-TMD 2 (m) | Reduction (%) | |
|---|---|---|---|---|---|
| | | | | EC-TMD 1 | EC-TMD 2 |
| Case 1 | 0.68 | 0.50 | 0.49 | 26.5 | 27.9 |
| Case 2 | 1.17 | 0.80 | 0.76 | 31.6 | 35.0 |
| Case 3 | 0.50 | 0.38 | 0.35 | 24.0 | 30.0 |
| Case 4 | 0.83 | 0.62 | 0.60 | 25.3 | 27.7 |
| Case 5 | 0.37 | 0.28 | 0.28 | 24.3 | 24.3 |
| Case 6 | 0.61 | 0.47 | 0.43 | 22.9 | 29.5 |

**Table 8.** RMS values of displacements at the tower top for different cases.

| Cases | No Damper (m) | With EC-TMD 1 (m) | With EC-TMD 2 (m) | Reduction (%) | |
|---|---|---|---|---|---|
| | | | | EC-TMD 1 | EC-TMD 2 |
| Case 1 | 0.25 | 0.21 | 0.20 | 16.0 | 20.0 |
| Case 2 | 0.43 | 0.32 | 0.31 | 25.6 | 27.9 |
| Case 3 | 0.18 | 0.15 | 0.14 | 16.7 | 22.2 |
| Case 4 | 0.30 | 0.24 | 0.23 | 20.0 | 23.3 |
| Case 5 | 0.14 | 0.12 | 0.11 | 14.3 | 21.4 |
| Case 6 | 0.23 | 0.18 | 0.17 | 21.7 | 26.1 |

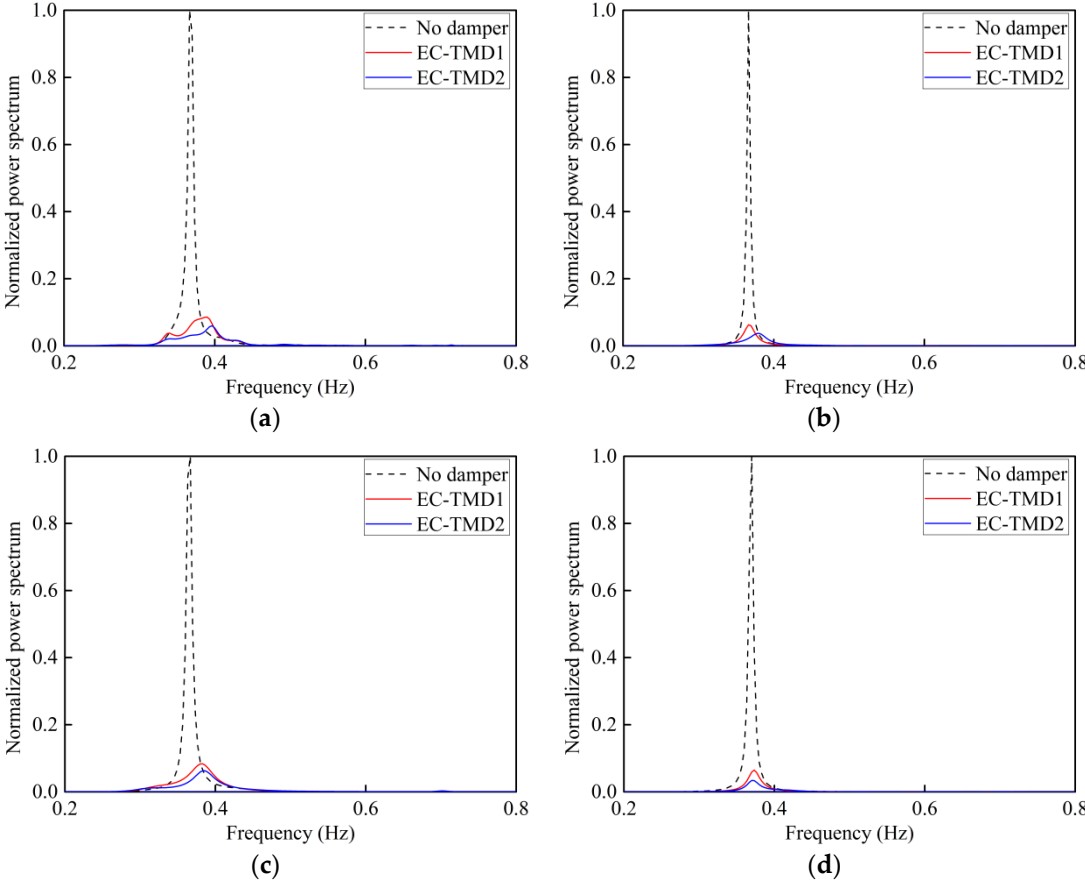

**Figure 17.** Acceleration power spectral density at the top of the tower with Class I and Class III: (**a**) Class I with 1 year's recurrence period; (**b**) Class I with 50 years' recurrence period; (**c**) Class III with 1 year's recurrence period; (**d**) Class III with 50 years' recurrence period.

## 6. Conclusions

The development of offshore wind power is an ideal choice for renewable and green energy supply. However, extreme winds pose a great threat to the safety of the OWT structures, which may result in structural failure and electric production reduction. Hence, the novel EC-TMD system is proposed to improve the vibration-reducing performance of the OWT structure. The main conclusions of this study can be summarized as follows:

(1) The EC-TMD system can combine the advantages of the ECD and TMD that are adjustable natural frequency and contactless damping. The damping mechanics of the EC-TMD system formulated from electromagnetic theory, which can be ideally treated as linear viscous damping characteristics.

(2) The EC-TMD system for vibration reduction is demonstrated by free attenuation and base-excitation tests. It can be observed that different gaps between the PMs and the copper plate as well as the PMs layout have great influences on the damping characteristic. Meanwhile, the coupling interactions between the excitation forces and the magnetic field forces apparently affect the damping features and vibration-reducing performance.

(3) To demonstrate the feasibility of the EC-TMD system in practical engineering, a 3D FEM is established to simulate the vibration responses of the OWTs supported by the CBF at the Xiangshui Wind Farm, together with the EC-TMD system installed on the top of the tower. Results show that the vibration displacement under extreme wind conditions can be mitigated by about 23% to 35% for peak values, and the acceleration can be reduced significantly.

However, some study limitations are also observable. On the one hand, even larger scale model tests should be conducted to further comprehensively investigate the damping features of the EC-TMD system, as in [13]. On the other hand, the EC-TMD system are recommended be applied in the future to practical engineering problems to verify its vibration-reducing performance.

**Author Contributions:** Conceptualization, J.L.; methodology, Y.Z. and C.L.; software, Y.Z. and C.L.; validation, H.W., and X.D.; formal analysis, Y.Z. and J.J.; resources, Q.J. and H.Z.; writing—original draft preparation, Y.Z.; writing—review and editing, X.D.; supervision, J.L., and H.W.; funding acquisition, J.L. and X.D.

**Funding:** This research was funded by the Innovation Method Fund of China, grant number 2016IM030100, the Fund for key research area innovation groups of China, grant number 2014RA4031, the Fund for National Natural Science Foundation of China, grant number 51709202, and the Tianjin Science and Technology Program, grant number 16PTGCCX00160.

**Acknowledgments:** The authors would like to express their gratitude to the respected editor and anonymous reviewers for their useful comments and precious time spent on this paper.

**Conflicts of Interest:** The authors declare no conflict of interest.

## Nomenclature

| | |
|---|---|
| OWT | Offshore wind turbine |
| EC-TMD | Eddy current with tuned mass damper |
| RMS | Root mean square |
| TMD | Tuned mass damper |
| TLD (TCLD) | Tuned (column) liquid damper |
| MRFD | Magneto-rheological fluid damper |
| ECD | Eddy current damper |
| CBF | Composite bucket foundation |
| PM | Permanent magnet |
| SDOF | Single degree of freedom |
| PSD | Power spectrum density |
| FEM | Finite element model |
| $B$ | Magnetic induction intensity |
| $J$ | Induced electric current density |
| $\sigma$ | Conductive coefficient |
| $F$ | Eddy current force |
| $D, H$ | Diameter, height of the CBF |
| $\bar{f}_{mean,i}, f_i(t), F_i(t)$ | Mean wind force, turbulent wind force and total wind force |

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
