# Peer review of "Application of an Eddy Current-Tuned Mass Damper to Vibration Mitigation of Offshore Wind Turbines"

_energies, doi:10.3390/en11123319_

Round 1

Reviewer 1 Report

MAIN COMMENTS The proposed work is interesting; the information provided seems to be valuable and useful for practical application. The authors present the results of various tests/cases and also some numerical simulation results. The manuscript is well structured. The title is too long and need to be changed with another shorter. After the Acknowledgments, a section denoted Nomenclature can be added because there are many abbreviations and notations considered in the work; this index of abbreviations would help in increasing the clarity of the proposed work. In the manuscript it is mentioned that a prototype was installed at Xiangshui Wind Farm. Do you have some measurements/tests at a real scale in order to be compared with the simulations? I did not see any comparison between measured and simulated data in order to validate your proposed methodology. Please explain better this aspect! Some minor corrections: Line 175: it is correct ‘system was investigated alone’ Line 222: it is better to write 2 mm Line 267: it is better ‘The CBF consists of a steel bucket’ Line 396 – in the title of the section correct - results

Author Response

Thank you for your precious time and useful commnets.

Point 1: The title is too long and need to be changed with another shorter. 

Response 1: In the revised manuscript, the title is changed to be “Application of an eddy current-tuned mass damper to vibration mitigation of offshore wind turbine” for comparatively simplify and easy to understand.

Point 2: After the Acknowledgments, a section denoted Nomenclature can be added because there are many abbreviations and notations considered in the work; this index of abbreviations would help in increasing the clarity of the proposed work.

Response 2: The authors would like the gratitude to the reviewer for the recommendation and corrections. In the revised manuscript, the abbreviations and some important notations are added in Nomenclature after the Acknowledgments, including 12 abbreviations and 9 notations. In this way, the index will help better understanding in this work.

Point 3: In the manuscript it is mentioned that a prototype was installed at Xiangshui Wind Farm. Do you have some measurements/tests at a real scale in order to be compared with the simulations? I did not see any comparison between measured and simulated data in order to validate your proposed methodology. Please explain better this aspect!

Response 3: In the revised manuscript, the following sentences as The first natural frequency of the model is 0.348 Hz to 0.35 Hz, which is almost the same as the measured values in Table 4. Therefore, the simulated model can be used to represent the real OWT structure.’  and the 2.0% damping ratio by prototype observation is adopted’ . Thus, both natural frequency and the damping ratio are almost the same for the simulated model and the real OWT structure, which is the comparison between measured and simulated OWT structure.

Point 4: Some minor corrections: Line 175: it is correct ‘system was investigated alone’ Line 222: it is better to write 2 mm Line 267: it is better ‘The CBF consists of a steel bucket’ Line 396 – in the title of the section correct – results.

Response 4: Some corrections are conducted based on the reviewer's comments in the new Lines of revised manuscript corresponding to the original file, including the ‘system was investigated alone’ in Line 175, ‘2 mm’ in Line 222, ‘The CBF consists of a steel bucket’ in Line 267, and “Results” in the title of the section in Line 400.

Reviewer 2 Report

The paper is well written; it deals with modeling and testing of an eddy current based mass damper for vibration mitigation of offshore wind turbine structures under extreme winds.

The article contains necessary background and motivation in the introduction section. Next, damping mechanics of the EC-TMD system and its implementation in analytical models are described in details; these are followed by small scale tests and numerical simulations.

The obtained results are discussed and conclude comprehensively.

Some minor remarks that can be taken into consideration:

1. Why in Table 1, there are two values for “Theoretical natural frequency (Hz)” column?

2. In line 448, in the sentence there should be probably “affect” instead of “effect”.

Author Response

Thank you for your precious time and useful commnets.

Point 1: Why in Table 1, there are two values for “Theoretical natural frequency (Hz)” column? 

Response 1: In original manuscript, the two values for “Theoretical natural frequency (Hz)” are the natural frequencies for the small-scale model in fore-aft and side-side directions, respectively. In the revised manuscript, only natural frequency in the fore-aft direction is reserved for avoiding misunderstanding.

Point 2: In line 448, in the sentence there should be probably “affect” instead of “effect”.

Response 2: Thanks for pointing out the mistake in the manuscript. In the revised manuscript, the “effect” is replaced by the “affect” in line 452. (in original one in Line 448).

Round 2

Reviewer 1 Report

The authors operated the modifications/explanations required.

Reviewer 2 Report

In the revised version of the article the Authors took into account the recommended amendments, thus making the work suitable for publication in Energies.